# S-D Net: Joint Segmentation and Diagnosis Revealing the Diagnostic Significance of Using Entire RNFL Thickness in Glaucoma

**Jie Wang**
Department of Automation
Tsinghua University
wangjie16@mails.tsinghua.edu.cn

**Chong Chen**
SenseTime Group Limited
chongbox@gmail.com

**Fei Li**
Zhongshan Ophthalmic Center
State Key Laboratory of Ophthalmology
Sun Yat-Sen University
lifei.aletheus@gmail.com

**Zhe Wang**
SenseTime Group Limited
wangzhe@sensetime.com

**Guoxiang Qu**
Shenzhen Institutes of Advanced Technology
Chinese Academy of Sciences
gx.qu@siat.ac.cn

**Yu Qiao**
Shenzhen Institutes of Advanced Technology
Chinese Academy of Sciences
yu.qiao@siat.ac.cn

**Hairong Lv**
Department of Automation
Tsinghua University
lvhairong@tsinghua.edu.cn

**Xiulan Zhang**
Zhongshan Ophthalmic Center
State Key Laboratory of Ophthalmology
Sun Yat-Sen University
zhangxl2@mail.sysu.edu.cn

## Abstract

Glaucoma is a severe eye disease causing blindness. The early diagnosis of glaucoma is of great importance and mainly based on the detection of early signs of optic neuropathy, including retinal nerve fiber layer (RNFL) thinning. We design S-D net to implement automatic segmentation of retinal layers in optical coherence tomography (OCT) images and diagnosis of glaucoma using RNFL thickness vector calculated from the segmentation maps. S-D net is an end-to-end optimized system that mimics the behavior of an ophthalmologist for diagnosing glaucoma with the OCT report. The specially designed unit layer in S-D net gives the threshold of thickness at each point of the RNFL in glaucoma diagnosis. Our results show that S-D net distinguishes glaucoma from healthy cases according to the distribution and magnitude of RNFL thickness. Our model achieves state-of-the-art segmentation results and competitive diagnosing accuracy compared with an experienced ophthalmologist.

## 1 Introduction

Glaucoma is the second leading cause of irreversible blindness [1], so the early detection of glaucoma is of great significance. Elevated intraocular pressure (IOP), visual field (VF) defect and glaucomatous optic neuropathy (GON) are the three main clinical elements for glaucoma diagnosis [23]. Retinal

1st Conference on Medical Imaging with Deep Learning (MIDL 2018), Amsterdam, The Netherlands.

nerve fiber layer (RNFL) thinning is the early signal of glaucoma. In clinical practice, the report of RNFL thickness by Optical Coherence Tomography (OCT) is widely used for diagnosis of glaucoma.

OCT is a rapidly developing technology and has made great impact on ophthalmology [23]. The advantages of OCT, such as fast scanning speed, non-invasiveness, high resolution and repeatability, make it widely used in eye diseases diagnosis. Glaucoma diagnosis makes use of OCT to measure the RNFL thickness of parapapillary regions [15]. Current clinical OCT machines give high-resolution cross sectional structure of retina, but rough and imprecise retinal sublayer segmentation (see Figure 1). Since the accurate segmentation of retinal layers is crucial in diagnosis and manual delineation is time-consuming, researchers have made great efforts to improve the performance of automatic segmentation. Graph based methods have been successfully applied in retinal layer segmentation [8, 7], including segmentation of retina with serous pigment epithelia detachments [21]. Based on recent developments in deep learning, He et al. [10] have proposed a modified U-shape fully convolutional network, i.e. S-Net, achieving the state-of-the-art in OCT retinal segmentation. Previous work [21, 10] involved dense preprocessing of OCT images and correction of segmentation results.

Current methods to diagnose glaucoma automatically often use either the vertical cup to disc ratio (CDR) or RNFL thickness to train a traditional classifier. Bechar et al. [1] proposed a semi-supervised learning method to segment cup and disc automatically to screen glaucoma. Xu et al. [25] combined 3D OCT data to get the final integrated RNFL thickness map. These two methods both adopted super pixel but not pixel-wise segmentation, which is inevitably incomprehensive. What is worse, noting that conventional RNFL thickness measurements in OCT reports are often summarized in 4 quadrants, 8 quadrants or 12 clock hours [25, 2], or in thickness maps [24], these sectoral measurements or maps lose information. It is necessary and critical for clinicians to have a comprehensive and precise view of RNFL thickness over the entire RNFL layer (each pixel in the width with a thickness value - number of pixels in the height) to make a decision about glaucoma (see Figure 1). However, quadrant-based RNFL thickness report remains the basis to diagnose glaucoma in clinical practice. We have found few research using the entire RNFL thickness vector to diagnose glaucoma, neither any deep understanding of the detailed RNFL thickness in diagnosis.

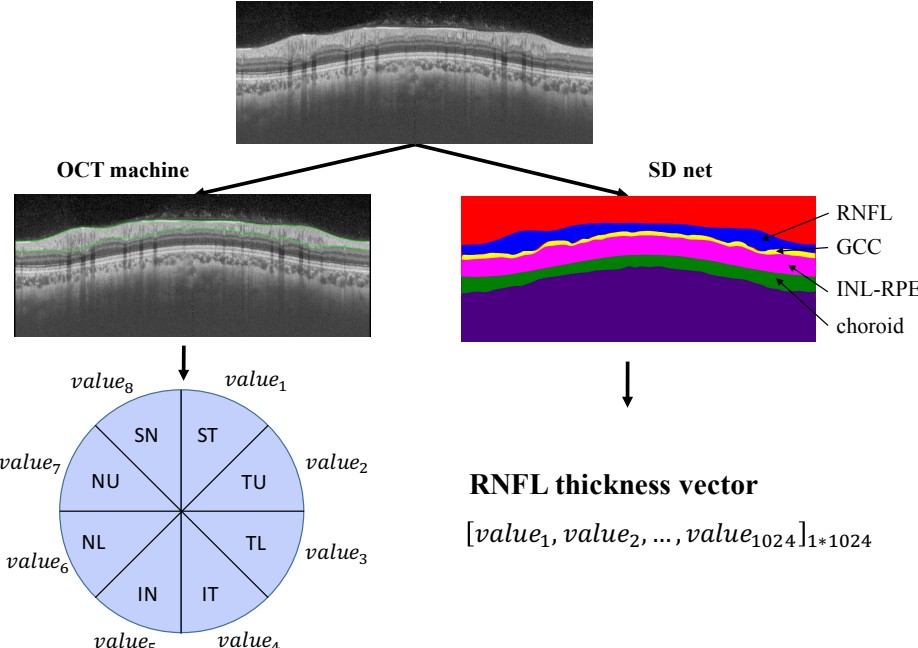

Figure 1: Conventional quadrant-based RNFL thickness obtained from OCT machine and entire RNFL thickness vector from our S-D net.

Here we propose a deep learning architecture, S-D net, to simultaneously segment retinal layers and diagnose glaucoma (see Figure 2). S-D net is composed of two parts, S-net for segmentation of retinal layers and D-net for diagnosis of glaucoma according to the entire RNFL thickness vector calculated from the segmentation results. S-net is inspired by the architecture designed by [10], which

is a modified U-net [19] with good performance in retina segmentation. Different from [10], our S-net accepts bigger input images without the pre-processing step of flattening or data augmentation and we use a different loss function for training. We carefully design D-net composed of several fully connected layers for the task of glaucoma diagnosis, in which the elaborate first layer tells the thickness threshold of RNFL in glaucoma diagnosis.

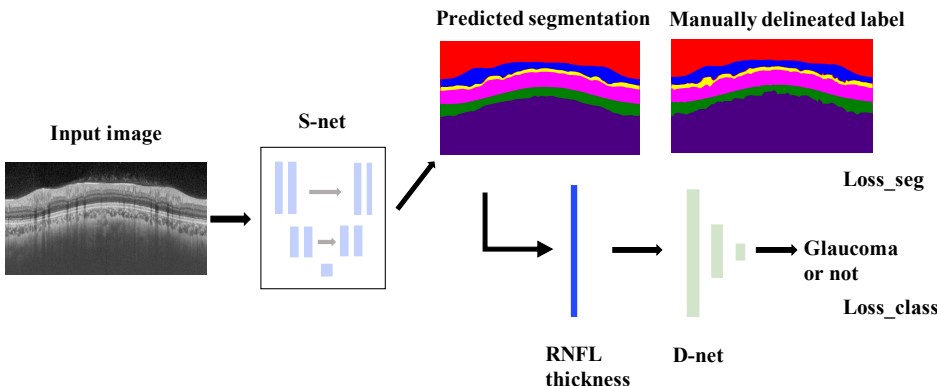

Figure 2: The overview of S-D net to segment OCT images and to diagnose glaucoma.

## 2 Related work

### 2.1 Retina segmentation

Current prominent automatic segmentation approaches in use for OCT images are 3-D graph based optimal surface segmentation methods [12, 17] proposed by Li el al. [13]. Garvin et al. [8] successfully applied the variations of graph search to normal retina segmentation by modeling the boundaries between retinal layers as terrain-like surfaces. Chen et al. [5] proposed probability constrained graph-search-graph-cut method to segment fluid-associated abnormalities in retinal OCT. Shi et al. [21] proposed multi-resolution graph search to segment retinal layers of OCT images with serous pigment epithelial detachments. They first preprocessed OCT images by denoising and alignment, and then detected retinal surfaces based on multi-resolution single surface graph search with correction and abnormal region with the help of the segmented layer surfaces.

Except for graph based methods, fully convolutional network (FCN) has shown state-of-the-art performance in pixel-wise labeling in semantic segmentation [14, 3, 4]. Convolutional network has strong representation ability of image features and shows state-of-the-art performance in many computer vision tasks such as image classification [11] and object detection [9, 20]. Fully convolutional network takes advantage of convolutional networks to complete the label predictions pixels-to-pixels. Ronneberger et al. [19] proposed a U-shape FCN called U-net to deal with very small training set, especially biomedical images. U-net has been successfully implemented in many tasks of biomedical images segmentation [6]. For OCT segmentation, He et al. [10] proposed a modified U-net called S-net and an identical topology correction net to improve the performance. They flattened OCT images and subdivided them into small overlapping patches to train the net. In this paper, we adopt the similar architecture of S-net. Different from previous work, our model accepts bigger inputs without alignment or subdividing and requires no correction after segmentation.

### 2.2 Glaucoma diagnosis

Current methods in automatic glaucoma diagnosis often adopt traditional machine learning methods, using either the vertical cup to disc ratio (CDR) or RNFL thickness as the diagnostic basis. Bechar et al. [1] proposed a semi-supervised learning method to complete the automatic cup and disc segmentation to screen glaucoma, in which the core parts are super pixel (small regions) segmentation, feature extraction and ensemble classification. Xu et al. [25] suggested that current sampling pattern – standard structural OCT along a 3.4 mm diameter circle centered at optic nerve head – is limited without taking full advantage of the 3D OCT images. The author combined 3D OCT data to get

the RNFL thickness map, then minimized blood vessel effect, and finally used ncut algorithm [22] to get the final processed RNFL thickness map. These two methods both adopt super pixel segmentation not pixels-to-pixels, which is inevitably incomprehensive. In addition, Muhammad et al. [16] used convolutional neural network to extract features from OCT images to diagnose glaucoma. Their research focuses on images and cannot explain the importance of RNFL thickness in the glaucoma diagnosis. More researches directly took conventional RNFL thickness measurements from OCT reports to train a classifier to diagnose glaucoma [25, 18]. Conventional RNFL thickness measurements are often summarized in 4 quadrants, 8 quadrants or 12 clock hours [25, 2], or in thickness maps [24], these sectoral measurements or maps lose information. In this paper, we develop S-D net to join retina segmentation and glaucoma diagnosis. We use the entire RNFL thickness vector instead of conventional quadrant-based thickness to train our model to diagnose glaucoma.

## 3  Architecture of S-D net

### 3.1  S-net

S-Net consists of a contracting path (left side) for feature extracting and a symmetric expansive path (right side) for pixel-wise labeling (See Figure 3). The contracting path takes a batch of 1024*560 images as input and uses two 3*3 convolutions (padded convolutions), each followed by a rectified linear unit (ReLU), batch normalization and a 2*2 max pooling operation with stride 2 for downsampling. After each downsampling we double the number of feature channels. There are four convolution-downsampling blocks in the contracting path. In the expansive path, each step consists of a 2*2 upsampling followed by the concatenation with the corresponding feature map from the contracting path and two 3*3 convolutions, each followed by a ReLU and batch normalization. The final layer uses a 1*1 convolution to map each 32-component feature vector to the desired number of classes. Here the final output is a 6*1024*560 volume corresponding to the probability maps for six parts: vitreous, RNFL, GCC, INL-RPE, choroid and sclera.

### 3.2  D-net

The ophthalmologist makes the diagnostic decision based on the report generated by the OCT machine, where the thickness of RNFL is represented by 8 quadrants or 12 clock hours, etc. In each quadrant, its color can be either green or red depending on its thickness compared to normal people. In order to mimic the behavior of the ophthalmologist and simultaneously to push it to the limit, we design D-net whose first layer is specially designed to try to understand the strategy of the network to do classification.

At the first layer called unit layer, the weight is an untrainable identity matrix, the bias is a trainable vector, and the input is a batch of RNFL thickness vectors with the dimension of 1024 calculated from the segmentation maps. In other words, this specially designed unit layer followed by ReLU can be formulated as Equation 1:

$$Output = \max(bias - Input * I, 0), \qquad (1)$$

where $I$ is the identity matrix. So the function of the unit layer equals that a trainable bias directly subtracts the input, and after training the learned bias can be expected as the thickness threshold to discriminate glaucoma and health. Here the unit layer is followed by two fully connected layers (see Figure 3), the former with a ReLU and the latter with a softmax to output the probability of each classification class. In this way, D-net is no longer a black-box classifier. We explicitly learn the bias in the first fully connected layer, which can be understood as the threshold at each location of RNFL. Once the measured thickness is smaller than the threshold, its signal is kept by ReLU and processed by later layers. And D-net takes all the 1024 thickness values into account instead of 4/8/12 quadrants. The threshold is thus automatically learned from massive training data, instead of tens of samples from the OCT machine [2]. Moreover, D-net can be jointly optimized with S-net, which makes S-D net an end-to-end trainable system.

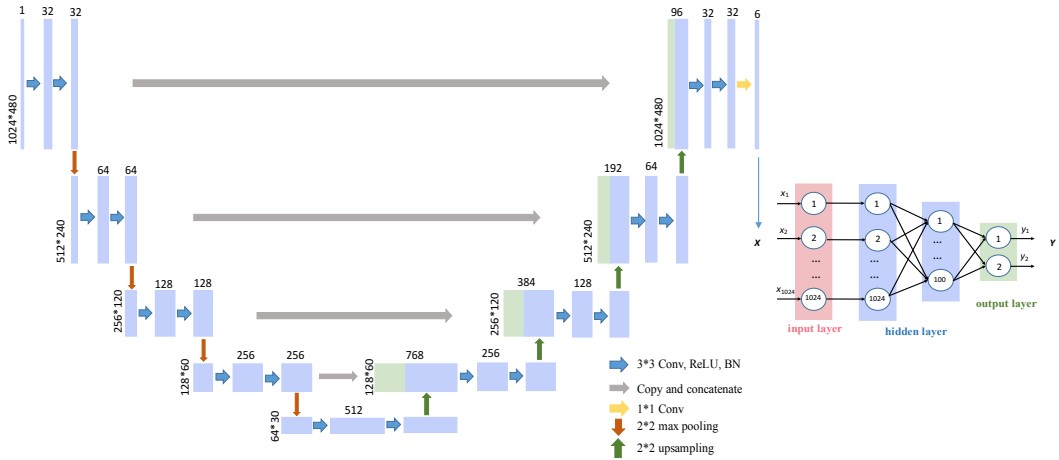

Figure 3: The architecture of S-D net.

## 4  Experiment

### 4.1  Data and preprocessing

Totally 1048 B-scan OCT images with disease labels from Zhongshan Ophthalmic Center of Sun Yat-sen University and the corresponding manually delineated segmentation maps (segmentation ground-truths) are enrolled in our study. We randomly split them into training set and testing set at an approximate proportion of 8:2. The training set consists of 517 glaucoma and 321 healthy cases, and the testing set has 128 glaucoma and 77 healthy cases. The ground-truth segmentation maps are labeled by professionals, where the RNFL, GCC (ganglion cell complex), and choroid are marked as red, green and blue, respectively. We transform these segmentation ground-truths into more precise and complete ones: the ground-truths with 3 labeled layers are transformed into equivalent ones with 6 discriminative labeled parts (see Figure 2). All the OCT images and ground-truth segmentations (1270 * 763) are first resized into 1024*763 and then safely cropped into 1024*560 remaining the core structure of retina.

### 4.2  Training

The OCT images with ground-truth disease labels and ground-truth segmentation maps in training set are used to train the S-D network implemented in TensorFlow. We adopt stochastic gradient descent to optimize the objective function. Our objective function combines two unweighted cross-entropy losses $Loss = Loss\_seg + Loss\_class$ where $Loss\_seg$ is for the pixel-wise dense segmentation, $Loss\_class$ is for the classification and each of them consists of a cross-entropy loss and a L2 regularization. More clearly, these two loss functions of S-D net are formulated as follows:

$$Loss\_seg = -\sum_{i=1}^{b}\sum_{j=1}^{h}\sum_{k=1}^{w}\sum_{m=1}^{c} y_{i,j,k,m}\log(\hat{y}_{i,j,k,m}) + \alpha \sum ||w\_1||_2, \qquad (2)$$

$$Loss\_class = -\sum_{i=1}^{b} y_i\log(\hat{y}_i) + \alpha \sum ||w\_2||_2, \qquad (3)$$

where $b$, $h$, $w$ and $c$ mean batch_size, image_height, image_width and image_channel respectively. $y_{i,j,k,m}$ denotes the ground-truth segmentation label, $\hat{y}_{i,j,k,m}$ denotes the predicted probability with that label of the pixel at height $j$, width $k$, feature channel $m$ and the $i$th sample, and $\alpha$ denotes the weight coefficient of L2 regularization; $y_i$ denotes the ground-truth classification label and $\hat{y}_i$ denotes the predicted probability of $i$th sample with the ground-truth label; ||w_1|| denotes all weights in S net and ||w_2|| denotes all weights in D net. We set the batch_size as 4, learning rate as 0.001, and $\alpha$ as

0.0005 during the training process on GeForce GTX 108. We must clarify that the data listed above have different proportions to the real retina structures because each OCT image has been resized and cropped differently in the process of manual delineation. To guarantee the thickness vector having the same proportion to the true thickness in the real world, we use unified original OCT images to get the thickness vectors on the basis of the pre-trained model. We first train S-D net end-to-end using the prepared data above, then we get thickness vectors of all unified original OCT images using the model from a checkpoint with good segmentation performance, and finally use the unified thickness vectors to fine tune the D net.

### 4.3 Evaluation metrics

We use three common metrics to evaluate the performance of segmentation: IoU (intersection over union), Dice, and F-score. We unify the definitions of all these three metrics according to TP (true positive), FP (false positive), TN (true negative) and FN (false negative):

$$IoU = \frac{TP}{TP + FP + FN}, \tag{4}$$

$$Dice = \frac{2 * TP}{2 * TP + FP + FN}, \tag{5}$$

$$Precision = \frac{TP}{TP + FP}, Recall = \frac{TP}{TP + FN}, F - score = \frac{2 * Precision * Recall}{Precision + Recall}. \tag{6}$$

In addition, we use accuracy, sensitivity, specificity and the average of these three metrics to evaluate the performance of classification.

### 4.4 S-D net in segmentation

We compare our best model trained end-to-end with other two state-of-the-art models on medical images segmentation. All the models adopt padded convolutions and are trained with the same data for the same epochs. Figure 4 shows samples of predicted segmentation maps in testing set using different models. The segmentation map of S-D net maintains the best physical topology of retina. Table 1 shows the quantitative segmentation results in average of retinal nerve fiber layer (RNFL) and all six layers in testing set. For RNFL our model achieves an average IoU of 92.3%, surpassing U-net by 28.8% and S-net by 1%. Also for both average Dice and average F-score, our model achieves 96.0%, surpassing U-net by 19.2% and S-net by 1%. For all six layers our model achieves an average IoU of 92.5%, surpassing U-net by 18.8% and S-net by 1%. For both average Dice and average F-score, our model performs the same well as S-net, achieving 95.9% and surpassing U-net by 12.9%.

Table 1: Segmentation results (RNFL in testing set)

| Model | RNFL | | | All layers | | |
|---|---|---|---|---|---|---|
| | IoU | Dice | F-score | IoU | Dice | F-score |
| U-net [19] | 0.635 | 0.768 | 0.768 | 0.737 | 0.830 | 0.830 |
| S-net [10] | 0.922 | 0.959 | 0.959 | 0.924 | 0.959 | 0.959 |
| S-D net (ours) | **0.923** | **0.960** | **0.960** | **0.925** | **0.959** | **0.959** |

### 4.5 S-D net in diagnosis

We used RNFL thickness vectors calculated from predicted segmentation maps to train our model to diagnose glaucoma. Our best model achieves an accuracy of 85.4%, sensitivity of 85.9%, specificity of 84.4% and and an average of 0.852 in testing data (see Table 2), mistaking 12 healthy cases as glaucoma and 18 glaucoma as healthy ones. For comparison, an experienced ophthalmologist diagnosed the same testing OCT images based on corresponding report images of RNFL thickness

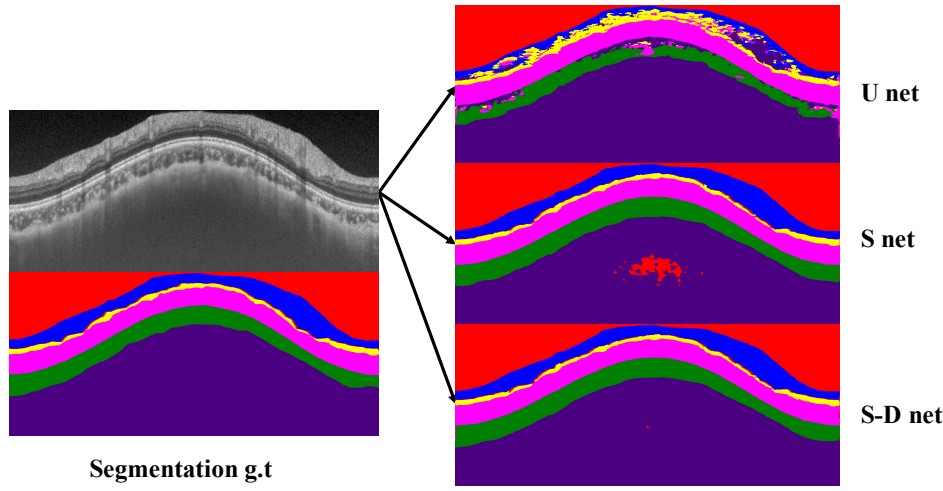

Figure 4: Segmentation maps predicted by different models of one sample in testing set.

Table 2: Diagnosing results (testing set)

| Name | ophthalmologist | S net - SVM | S-D net | | |
|---|---|---|---|---|---|
| RNFL Thickness | 1*1024 | 1*1024 | 1*1024 | 1*5 | 1*8 |
| Accuracy | 0.566 | 0.688 | **0.854** | 0.756 | 0.771 |
| Sensitivity | 0.313 | 1.000 | 0.859 | 0.750 | 0.797 |
| Specificity | 0.987 | 0.169 | 0.844 | 0.766 | 0.727 |
| Average | 0.622 | 0.619 | **0.852** | 0.757 | 0.765 |

(see Figure 5(a)). In report images, the shallow green region presents the range of healthy RNFL thickness at each point of the 1*1024 vector from training set and the red line presents the thickness vector of the current sample to be evaluated in testing set. The diagnosis of the doctor reaches an accuracy of 56.6%, sensitivity of 31.3%, specificity of 98.7% and an average of 0.622, mistaking 1 healthy case as glaucoma but 88 glaucoma as healthy. Compared with the doctor's diagnosis, S-D net shows much higher accuracy and sensitivity, and little weaker specificity on distinguishing an OCT image to be glaucoma or not based only on the RNFL thickness vector. In addition, we changed D net into a SVM classifier which reaches an accuracy of 68.8%, sensitivity of 100%, specificity of 16.9% and an average of 0.619 in testing set, finding all glaucoma but mistaking 64 healthy cases as glaucoma. D net outperforms SVM on classification accuracy and specificity with a big margin. Altogether S-D net shows the best performance on diagnosing glaucoma.

## 4.6 Analysis of S-D net strategies

We analyzed the average RNFL thickness vector in true disease, true health, false disease and false health group. Figure 5(b) shows the average thickness of these four groups in blue, orange, green and red lines. In training set, blue line (true disease) with low magnitude separates from orange line (true health) with high magnitude discriminatively, and most parts of red line (false health) are higher than orange line. Similar situations occur in testing set. We suggest that S-D net discriminates glaucoma and healthy case according to the distribution and magnitude of RNFL thickness. We also note that some healthy people have thin RNFL while some patients with glaucoma have no RNFL thinning, bringing difficulty to the S-D net. As we mentioned above, the unit layer is designed and expected to tell the thickness threshold with its bias. Our experiments verified this assumption. We initialized bias in different ways, including a normal distribution with mean 0 (s.d. = 1), a normal distribution with mean 19 equal to the average value of thickness vector in training data (s.d. = 1), and the average thickness vector of training data. We found the initial setting as the average thickness vector achieves the best. We note that the bias is difficult to update during training.

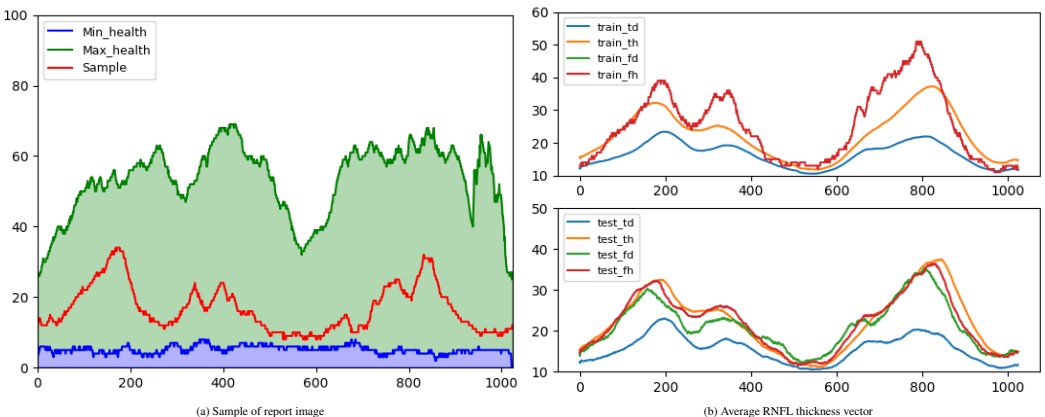

<p>(a) Sample of report image        (b) Average RNFL thickness vector</p>

Figure 5: (a): Sample of report image where shallow green region presents the range of healthy RNFL thickness from training set and the red line presents the thickness vector of the current sample to be evaluated in testing set. (b): Average RNFL thickness vector of four predicted groups – true-disease, true-health, false-disease, and false-health – in training set and testing set.

Further, we found some regions of RNFL are more critical than others in the thickness vector in glaucoma diagnosis. We evaluated the classification performance of the thickness vector lacking a portion of 100 consecutive data points by covering the region sequentially from 0 to 1024. The results show that when covering regions 0:100, -400:-300, -300:-200, -200:-100 and -100:0, the classification accuracy declines obviously. From a doctor's view, these five regions correspond to region near the temporal quadrant where glaucomatous optic neuropathy usually starts. Also, we compared the classification results using the entire RNFL thickness vector, the average thickness of the above five regions and conventional eight quadrants in clinical practice. Results show that using the entire RNFL thickness vector is much better than the other two methods, and using the average thickness of conventional eight quadrants is better than using that of the five regions. Using five regions achieves accuracy of 75.6%, sensitivity of 75.0%, specificity of 76.6% and an average of 0.757. And using eight regions achieves accuracy of 77.1%, sensitivity of 79.7%, specificity of 72.7% and an average of 0.765. These results demonstrate that taking use of the entire RNFL thickness is of great importance in glaucoma diagnosis. (see Table 2).

Finally, we found that counting other layers's thickness in makes no improvement in glaucoma diagnosis. We conducted three experiments using RNFL thickness vector, RNFL+GCC thickness vector and RNFL+GCC+INL-RPE thickness vector respectively. Results show that using RNFL thickness vector is much better than using RNFL+GCC thickness vector or using RNFL+GCC+INL-RPE thickness vector. Using RNFL + GCC achieves accuracy of 72.2%, sensitivity of 64.1% and specificity of 85.7%. And using RNFL+GCC+INL-RPE achieves accuracy of 62.4%, just taking all testing images as glaucoma.

## 5 Conclusion

We propose an end-to-end trained S-D net to simultaneously implement automatic segmentation of OCT images and diagnosis of glaucoma. Our S-D net achieves state-of-the-art segmentation performance in OCT images and much better performance in glaucoma diagnosis than an experienced ophthalmologist. The classification accuracy using RNFL thickness vector calculated from segmentation maps also exceeds that using conventional quadrant-based RNFL thickness. Considering that glaucoma is a complex eye disease and the diagnosis involves multiple factors such as IOP, VF test, CDR and RNFL thinning in clinical practice, we conclude that our method of automatic glaucoma diagnosis based only on RNFL thickness vector has achieved a significant success. And we suggest that using the entire RNFL thickness is more precise than using the conventional quadrant-based RNFL thickness in glaucoma diagnosis.

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
