# OpenReview forum: "S-D Net: Joint Segmentation and Diagnosis Revealing the Diagnostic Significance of Using Entire RNFL Thickness in Glaucoma"
_MIDL.amsterdam/2018/Conference — Submitted to MIDL 2018_

### Review · AnonReviewer1 · 2018-05-04
**paper well written, simple model, no state-of-the-art benchmark**

**Rating:** 3
**Confidence:** 2

**Review:**

This paper tackles the relevant problem of glaucoma diagnosis by means of RNFL segmentation. Authors show that assisting glaucoma diagnosis with segmentation information improves some baselines.

pros

+ the paper is well written and easy to follow
+ simple model trained end-to-end using segmentation as auxiliary task to diagnosis

cons
- no state-of-the-art benchmark

The paper is well written and easy to follow.  It seems reasonable to build an end-to-end model to segment the retinal nerve fiber layer as an auxiliary task to compute its thickness and facilitate glaucoma diagnosis. However, I have some concern related to the baseline and a few comments/questions regarding the experimental setup:

*What kind of upsampling is used in the expansive path of the S-Net? (transposed convolution, nearest neighbor, etc)
*Is it necessary to segment 6 classes? It seems that only RNFL is useful to diagnose glaucoma (according to the experimental results). Why not segment RNFL only?
*In the unit layer, is one single bias learnt, or one bias per position?
*Authors claim that the bias serves as thickness threshold to discriminate glaucoma from healthy subjects. It might be worth exploring the biases learnt by the final model and analyze how they relate to the diagnosis decisions.
*How important is the segmentation step to diagnose glaucoma? It would be beneficial to include a baseline that tries too predict glaucoma directly from the input of the S-net.
*The way to calculate the thickness vectors is not described in the paper. Is is just the sum of pixels predicted as RNFL?
*D-net has 2 possible outputs, why not use a single neuron as D-net output instead of 2 (as shown in Figure 3)?
*How are the random train/valid/test splits performed? Is there any subject overlap among different sets?
*How are the ground truth labels transformed from 3 to 6 labels?
*Why resize and crop images instead of using the full resolution ones?
*In the losses, is the alpha term different for w_1 and w_2? Why not use a single regularization term for the whole S-D-Net?
*Would results change if the cross entropy was weighted following class imbalance?
*It seems from section 4.2. that D-Net is finetuned with full resolution images, why not finetune the whole S-D net?
*Metrics: I don't find it necessary to report both IoU and Dice coefficient, since one can be computed from the other and they both measure overlap between segmentation and ground truth.
*Is the dataset used publicly available?
*Are all models trained until convergence? It seems that the only criteria is "for the same number of epochs". Are the models early stopped based on validation performance?
*If you segment 6 classes, why not report the results for all of them, and compare to other models to understand which models are better at segmenting which classes? (It is still unclear to me why one would want to segment 6 classes in this context).
*Comparison to U-Net out-of-the-box, whose architectural and optimization hyper-parameters were not tuned to address the same problem seems a bit unfair. It might be interesting spending some time tuning U-Net hyperparameters to have a better comparison.
*On table 1, it seems that the difference between S-net and S-D-net is 0.1%.
*Since the text report results in %, it might be more consistent to report results in table 1 and 2 as % as well.
*Reporting accuracy, specificity and sensitivity is enough, there is no need to average those metrics.
*Is the S-Net-SVM in table 2 trained by finetuning S-Net when training the SVM?
*Could the authors provide some insights on the difficulty of training biases?
*It might be worth incorporating additional cues (other than thickness) to the pipeline to see how performance changes.
*Please add tensorflow reference.
*It would be beneficial to report results on a publicly available dataset benchmarked by other state-of-the-art models.


**Special Issue:**

No

---

> ### Comment · ~Jie_Wang3 · 2018-05-13
> **Reply to reviewer1 - 1**
>
> *What kind of upsampling is used in the expansive path of the S-Net? (transposed convolution, nearest neighbor, etc)
> —— We use transposed convolution with bilinear interpolation weights.
>
> “Is it necessary to segment 6 classes? It seems that only RNFL is useful to diagnose glaucoma (according to the experimental results). Why not segment RNFL only?”
> ——According to the manual delineation, segmenting 6 layers are the most suitable in our data. We have done experiments using more layers to predict glaucoma (see P8), and results show that only RNFL is useful to diagnose glaucoma. But this conclusion is drawn after the experiments. So we cannot segment RNFL only at the beginning. What’s more, even though only RNFL is useful to diagnose glaucoma, in clinical practice, segmenting other layers is useful for doctors to diagnose other diseases.
>
> “In the unit layer, is one single bias learnt, or one bias per position?”
> ——One bias per position.
>
> “Authors claim that the bias serves as thickness threshold to discriminate glaucoma from healthy subjects. It might be worth exploring the biases learnt by the final model and analyze how they relate to the diagnosis decisions.”
> ——Exactly. As we mentioned in the paper: the bias is difficult to update during training. That the bias serves as thickness threshold can be validated from the fact that when we initialize the bias as the average thickness value of the training set at each position, we get the best classification result. And when we initialize the bias randomly, the classification performance drop.
>
> “How important is the segmentation step to diagnose glaucoma? It would be beneficial to include a baseline that tries to predict glaucoma directly from the input of the S-net.”
> ——As for our paper, our purpose is to clarify the significance of RNFL thickness in diagnosing glaucoma. Since in clinical practice doctors use the quadrant-based RNFL thickness when diagnosing glaucoma, we argue this is partial. We use the RNFL thickness vector including every position of RNFL to diagnose glaucoma and results show that using entire RNFL thickness is more accurate than conventional quadrant-based thickness. So the segmentation step is the base of our purpose, and we must get thickness first by segmentation. We added this experiment (please see the second last response for Reviewer 2.) even though it seems to have little relation with our purpose.
>
> “*The way to calculate the thickness vectors is not described in the paper. Is is just the sum of pixels predicted as RNFL?”
> ——Your understanding is likely to be right. As we mentioned in the paper: each pixel in the width with a thickness value - number of pixels in the height. We calculate the thickness vectors according to this, i.e., thickness value at the each position of width is the sum of pixels in the height predicted as RNFL.
>
> *D-net has 2 possible outputs, why not use a single neuron as D-net output instead of 2 (as shown in Figure 3)?
> —— Because we use soft-max cross-entropy loss function instead of sigmoid. They are equivalent in theory.
>
> *How are the random train/valid/test splits performed? Is there any subject overlap among different sets?
> ——We randomly split all images into train/test by 8:2 without consideration of subject. Because one subject has at most two images (left eye and right eye), and these two images can be regarded as independent in glaucoma.
>
> *How are the ground truth labels transformed from 3 to 6 labels?
> —— As mentioned in paper, only RNFL, GCC, and choroid are delineated by professionals. According to the manual delineation, 3 delineated layers equal to 6 labeled layers in OCT images: layer above RNFL, RNFL, GCC, layer between GCC and choroid, choroid, and layer below choroid (see Figure 1). To put it more clearly, an image has six layers, and the second, the third and the fifth are labeled by 1, 2, 3; then we re-label all the six layers by 1,2,3,4,5,6 from the first to the sixth layer.
>
> *Why resize and crop images instead of using the full resolution ones?
> —— The full resolution images are too large for the network to process and contain a large portion of meaningless background. Meanwhile, we also need to ensure the thickness vectors of different images are under the same scale, so we safely crop the images and resize them to have the same scale.
>
> *In the losses, is the alpha term different for w_1 and w_2? Why not use a single regularization term for the whole S-D-Net?
> —— Actually they are the same. We use a single regularization term in experiments.

---

> ### Comment · ~Jie_Wang3 · 2018-05-14
> **Reply to reviewer1 - 2**
>
> *Would results change if the cross entropy was weighted following class imbalance?
> —— Thanks for your suggestion. Considering that the data has no severe problem of class imbalance, we do not use weighted cross entropy in our original paper. Now we have tried weighted cross entropy loss, glaucoma cases are weighted as 0.4 = |N|/(|N| + |p|) and healthy cases are weighted as 0.6 = |P|/(|N| + |p|), where |N| is the number of healthy cases and |P| is the number of glaucoma cases. The classification accuracy does not change. We also try other weights, but accuracy drops. There are a little more glaucoma cases than healthy ones, which is a good thing because it is common that disease case is hard to be distinguished from healthy ones. So in our paper, we suggest there is no need to use weighted cross entropy to balance class.
>
> *Metrics: I don't find it necessary to report both IoU and Dice coefficient, since one can be computed from the other and they both measure overlap between segmentation and ground truth.
> —— Equation 4 and equation 5 denote IoU and Dice, and it seems that one cannot be computed from the other. Reference 19 use IoU, and reference 10 use Dice, so we show these two metrics both even though they both measure overlap between segmentation and ground truth.
>
> *Is the dataset used publicly available?
> —— Now the dataset is not publicly available. We will public it soon.
>
> *Are all models trained until convergence? It seems that the only criteria is "for the same number of epochs". Are the models early stopped based on validation performance?
> ——In the comparison experiments, all models are trained for the same number of epochs, and actually they all converge when stopping training.
>
> *If you segment 6 classes, why not report the results for all of them, and compare to other models to understand which models are better at segmenting which classes? (It is still unclear to me why one would want to segment 6 classes in this context).
> ——As shown in Table 1, we report the average segmentation results of 6 layers. Again, we have done experiments using more layers to predict glaucoma (see P8), and results show that using RNFL only has higher accuracy in diagnosing glaucoma. But this conclusion is drawn after the experiments. So we cannot segment RNFL only at the beginning. What’s more, even though only RNFL is useful to diagnose glaucoma, in clinical practice, segmenting other layers is useful for doctors to diagnose other diseases.
>
>
> *Comparison to U-Net out-of-the-box, whose architectural and optimization hyper-parameters were not tuned to address the same problem seems a bit unfair. It might be interesting spending some time tuning U-Net hyperparameters to have a better comparison.
> —— Right. Actually S-net [10] is just the architectural optimized version of U-net [19]. So we do not find it necessary to further tune the parameters of U-Net,  which is out of the scope of this paper. Again, we have to emphasize our purpose: we proposed an interpretable network for diagnosing glaucoma by explicitly learning the thickness of RNFL like a doctor. We think an acceptable segmentation network already meets our demands.
>
> *On table 1, it seems that the difference between S-net and S-D-net is 0.1%.
> —— Right. Thanks for your carefulness, and we will correct this typo.
>
> *Since the text report results in %, it might be more consistent to report results in table 1 and 2 as % as well.
> —— Right. Thanks for your suggestion. We will unify them.
>
> *Reporting accuracy, specificity and sensitivity is enough, there is no need to average those metrics.
> —— Thanks for your suggestion.
>
> *Is the S-Net-SVM in table 2 trained by finetuning S-Net when training the SVM?
> ——S-net is fixed.
>
>  *Could the authors provide some insights on the difficulty of training biases?
> —— As we mentioned in paper (see P7, section 4.6), the initialization of biases affects the results a lot. So carefully select the initialization of bias.
>
> *It might be worth incorporating additional cues (other than thickness) to the pipeline to see how performance changes.
> —— Yes, this is a good suggestion and this is important in clinical practice. Next paper we will consider this.
>
> *Please add tensorflow reference.
> —— Thanks. We will add it.
>
> *It would be beneficial to report results on a publicly available dataset benchmarked by other state-of-the-art models.
> —— It seems that there is no publicly available OCT dataset with both segmentation labels and glaucoma labels.

---

### Review · AnonReviewer2 · 2018-05-09
**Good results, but experimental details are not clear**

**Rating:** 2
**Confidence:** 2

**Review:**

This work presents a pipeline for the diagnosis of glaucoma in OCT images. First, a U-net based model is used to segment the RNFL, which is then transformed into a thickness vector, and is then passed through a fully-connected net to diagnose glaucoma. Results seem promising, but the specifics of the experiments are unclear, making it difficult to determine if a robust validation was conducted.

Pros:
- Flow is easy to follow, but several points still require clarification
- Thorough literature review
- Figure 1 is very informative and a good representation of the task being solved
- Simple models, easy to understand

Cons:
- No standard deviation reported in results
- Needs significant grammar revisions
- "Thickness vector" term not clearly define anywhere
- Details of experiments and validation are unclear
- In Table 1: S-net and the proposed model achieved nearly identical results (e.g. a 0.1% improvement). The similarity in these results should be discussed, instead of glazing over it and generalizing by saying that the S-D net has the best performance. In a statistical analysis, it's likely that this isn't a significant improvement - but standard deviation isn't reported so it's impossible for a reader to make any conclusions regarding this.
- "For RNFL our model achieves an average IoU of 92.3%, surpassing U-net by 28.8% and S-net by 1%." - No, its 0.1%

Comments:
- "We explicitly learn the bias in the first fully connected layer, which can be understood as the threshold at each location of
RNFL." What does this threshold mean? The definition of "thickness" vector is not well-defined and it's difficult to understand what is going on here.
- Figure 5: axes not labelled
- Why not use any pre-processing or data augmentation on the images?
- Evaluation Metrics section: F-score equation requires formatting (change minus to hyphen)
- Results reported as both percentages and decimals - choose one and be consistent
- Why were 6 classes segmented when only one was used for analysis?
- How were the ground truth images transformed from 3 to 6 labels?
- How many experts made the delineations?
- On validating diagnostic accuracy: How were the diagnostic ground truth labels generated? Were they labelled by an expert? If so, then maybe the ophthalmologist results in Table 2 just indicate low agreement between two observers. The details here are too unclear for one to draw a conclusion.
- Table 1: Were the U-Net and S-Net optimized for this dataset? Need more details here
- Why not test to see if an accurate diagnosis can be generated without the intermediate step of RNFL segmentation? i.e. a model that takes the image data and just generates a diagnosis from that?
- Why not test against another deep learning model instead of a SVM? There's room for a lot more analysis here, considering that there's no real benchmark

Overall, this results are promising, but more details and experiments are required.

**Special Issue:**

No

---

> ### Comment · ~Jie_Wang3 · 2018-05-13
> **Reply to reviewer2 - 1**
>
> ”No standard deviation reported in results”
> —— We trained S-D Net from scratch for ten times and obtained the standard deviation of mean IoU (0.0033), Dice (0.0018) and F-score (0.0018), respectively.
>
> "Thickness vector" term not clearly define anywhere
> —— As mentioned in Introduction: each pixel in the width with a thickness value - number of pixels in the height. So, as for all the 1024*560 images fed into the network, the thickness vector is 1*1024.
>
> “In Table 1:….In a statistical analysis, it's likely that this isn't a significant improvement - but standard deviation isn't reported so it's impossible for a reader to make any conclusions regarding this.”
> —— Yeah, we acknowledge S-D net just achieves a competitive segmentation performance to the state-of-the-art. But again we have to claim that our purpose is to make more accurate glaucoma diagnosis and reveal the significance of using the entire RNFL thickness to diagnose glaucoma based on the RNFL thickness from the automatic segmentation results with an acceptable accuracy. So the just competitive segmentation results are already suitable for the task of diagnosis.
>
> "For RNFL our model achieves an average IoU of 92.3%, surpassing U-net by 28.8% and S-net by 1%." - No, its 0.1%
> —— Right. Thanks for your carefulness, and we will correct this typo.
>
> "We explicitly learn the bias in the first fully connected layer, which can be understood as the threshold at each location of RNFL." What does this threshold mean? The definition of "thickness" vector is not well-defined and it's difficult to understand what is going on here.
> —— Retina is a physiological structure of eye, and it does have thickness. Then in the OCT image, as mentioned in paper: each pixel in the width with a thickness value - number of pixels in the height. So for the 1024*560 images fed into the network, the thickness vector is 1*1024. As mentioned in the paper, RNFL thinning is an early sign of glaucoma. Ophthalmologists verify the OCT report by comparing the RNFL thickness to those of normal people when diagnosing glaucoma. We mimic this process by explicitly learning the thickness threshold in a data driven way. In this way, we can interpret what the network has learned and how the thickness vector affect the diagnosis results.

---

> ### Comment · ~Jie_Wang3 · 2018-05-14
> **Reply to reviewer2 - 2**
>
> “Figure 5: axes not labelled”
> —— X axis means the 1*1024 thickness vector, and y axis means thickness (measured in number of pixels).
>
> “Why not use any pre-processing or data augmentation on the images?”
> —— We tried multi-scale training but observed no significant improvement. Maybe it is because the images used to train the S-D net have already been randomly scaled and cropped.  Besides, we require the thickness to be in the same scale, so in the second stage, we do not want to do any pre-processing that may influence the scale of the image.
>
> “Evaluation Metrics section: F-score equation requires formatting (change minus to hyphen)”
> —— Thanks. We will correct this.
>
> “Results reported as both percentages and decimals - choose one and be consistent”
> —— Thanks. We will correct this.
>
> “Why were 6 classes segmented when only one was used for analysis?”
> —— First we use RNFL only because RNFL thinning is most related to glaucoma in clinical practice. Second, segmenting 6 layers is to train a segmentation network. Third, we do experiments using more layers as mentioned in P8, and results show that using RNFL only is suitable for glaucoma diagnosis.
>
> “How were the ground truth images transformed from 3 to 6 labels? ”
> —— As mentioned in paper, only RNFL, GCC, and choroid are delineated by professionals.  Further, according to the manual delineation, we get 6 equivalent layers: layer above RNFL, RNFL, GCC, layer between GCC and choroid, choroid, and layer below choroid (see Figure 1). More specifically, an image has six layers, and the second, the third and the fifth are labeled by professionals as 1, 2, 3; then we re-label all the six layers by 1,2,3,4,5,6 from the first to the sixth layer.
>
>  “How many experts made the delineations?”
> —— Three
>
> “On validating diagnostic accuracy: How were the diagnostic ground truth labels generated? ”
> —— The ground truth labels are the real clinical diagnostic results.
>
>  “Were they labelled by an expert? “
> —— Yes.
>
> “Table 1: Were the U-Net and S-Net optimized for this dataset?”
> —— We train U-Net and S-Net from scratch using our dataset, and when stopping training, they both converge.
>
> “Why not test to see if an accurate diagnosis can be generated without the intermediate step of RNFL segmentation? i.e. a model that takes the image data and just generates a diagnosis from that?”
> —— We trained a VGG network that takes in the OCT images as input and directly predicts the diagnosis result. Its performance is slightly worse than S-D Net (accuracy = 0.845). However, this network is a black box and we know little about what it learns. On the contrary, our purpose is to mimic the diagnosis process of a doctor and try to understand the significance of RNFL thickness in diagnosing glaucoma.
>
> “Why not test against another deep learning model instead of a SVM?”
>  —— S-D Net itself can be seen as two deep learning models, i.e., one for segmenting retinal layers and the other for diagnosing glauscoma. We do not see the point of an additional experiment with another deep learning model. Meanwhile, SVM is a traditional classifier achieving good performance in classification before deep learning. So we compare deep learning model with SVM.

---

### Review · AnonReviewer3 · 2018-05-09
**Interesting results and approach, details and writing unclear.**

**Rating:** 2
**Confidence:** 2

**Review:**

The paper presents neural network pipeline for joint segmentation and diagnosis of Glaucoma from retinal OCT images. It combines a segmentation network S-net and an interpretable diagnostic network, which uses the thicknesses output from the S-net to diagnose Glaucoma. It is tested on data from 128+77 glaucoma and healthy subjects. Evaluating segmentation performance with respect to manual annotations and a diagnostic performance with respect to an undefined ground truth.

Pros:
- The diagnostic performance appears to be a good deal better than the doctor's performance.
- The interpretability of the diagnostic network makes it possible for the authors to investigate what measurements matter for diagnostic accuracy.

Cons:
- The segmentations results do not appear to be significantly different from prior work.
- The writing could be improved and made more succinct, and some important details of the methods and evaluation are unclear.

Specific comments:
S-net is inspired by reference 10, however the differences are not very clear from the work. Table 1 shows that there is very little, if any, difference in segmentation quality.

The experience level of the doctor is not given.

Related prior work is discussed both in the Introduction and section 2 with a lot of repeating information.

P3, "Different from previous work, our model accepts bigger inputs without alignment or subdividing and requires no correction after segmentation" (a very similar statement can be found towards the bottom of the page), not clear what this means exactly, what makes it possible, and what advantages it might carry.

P4, "then minimized blood vessel effect" what is this?

The description of the data does not cover how the diagnostic groundtruth was established.

"We set the batch_size as 4, learning rate as 0.001, and alpha as 0.0005", how were these hyper parameters set?

"We transform these segmentation ground-truths into more precise and complete ones: the ground-truths with 3 labeled layers are transformed into equivalent ones with 6 discriminative labeled parts", how?

What do you mean by: "We use unified original OCT images to get the thickness vectors on the basis of the pre-trained model."?

Why first train the S-D net end-to-end and then afterwards tune the D net?

The percentage improvements listed in section 4.4 are percentage points improvements and the improvements listed as 1 % are actually 0.1 pp.

I see no reason to list averages of accuracy, sensitivity and specificity.

The comparison SVM approach is not described in adequate detail.

P7, "We note that the bias is difficult to update during training.", what is the point?

On P8 an experiment is described which investigates what removing parts of the thickness vector does to the diagnostic ability. How should the regions listed, "0:100, -400:-300, -300:-200, -200:-100 and -100:0" be interpreted and what was put instead of these values, as a constant thickness vector length of 1024 seems to be assumed. An experiment is also conducted on including thickness measurements of other layers, but the specifics of how this is done could be more clear. E.g. is the thickness vector extended with the information of the extra layers?

Minor comments:
Sentences (and parts of sentences) which do not make sense:
  P2, "... which is inevitably incomprehensive" (also on P4)
  P2, "(each pixel in the width with a thickness value - number of pixels in the height)"
  P2, "neither any deep understanding of the detailed RNFL thickness in diagnosis"
  P4, "So the function of the unit layer equals that a trainable bias directly subtracts the input, and after training the learned bias can be expected as the thickness threshold to discriminate glaucoma and health."
  P6, "Table 1 shows the quantitative segmentation results in average of retinal nerve fiber layer (RNFL) and all six layers in testing set."
  P6, "Table 1 shows the quantitative segmentation results in average of retinal nerve fiber layer (RNFL) and all six layers in testing set."

P2, "fiber layer (RNFL) thinning is the early signal of glaucoma" -> "fiber layer (RNFL) thinning is an early sign of glaucoma"
P2, "few research" -> "little research"
P3, "Convolutional network has strong representation..." and "Fully convolutional network takes advantage", missing either article or plural. This seems to be a common problem throughout the manuscript.
P3, "In this paper, we adopt the similar architecture of S-net", -> "... a similar architecture as..."
P4, "More researches" -> "Other studies"
P5, "remaining the core structure" -> "keeping the core structure"
P5, "where b, h, w and c mean batch_size", "mean" -> "denotes"
P6, "end-to-end with other two state-of-the-art", "other two" -> "two other"
p8, "Finally, we found that counting other layers’s thickness in makes no improvement in glaucoma diagnosis.", "counting other layers's thickness in" -> "including other layers' thickness"



**Special Issue:**

No

---

> ### Comment · ~Jie_Wang3 · 2018-05-13
> **Reply to reviewer3 - 1**
>
> “The segmentations results do not appear to be significantly different from prior work.”
> ——We acknowledge retinal segmentation results are just competitive to the state-of-the-art. But we have to claim that our purpose is to make more accurate glaucoma diagnosis and reveal the significance of using the whole RNFL thickness to diagnose glaucoma based on the RNFL thickness from the automatic segmentation results with an acceptable accuracy. So the just competitive segmentation results are already suitable for the task of diagnosis.
>
> “The experience level of the doctor is not given.”
> ——The doctor has 5-year clinical experience.
>
> P3, "Different from previous work, our model accepts bigger inputs without alignment or subdividing and requires no correction after segmentation" (a very similar statement can be found towards the bottom of the page), not clear what this means exactly, what makes it possible, and what advantages it might carry.
> ——Because we safely crop the OCT images into 1024*560 which can be fed into the network directly.  The advantages are simplifying the process of OCT images’ segmentation, getting the thickness vectors from the segmentation results directly and so on.
>
>  P4, "then minimized blood vessel effect" what is this?
> ——This means reference 25 does some pre-processing on OCT images.
>
> “The description of the data does not cover how the diagnostic groundtruth was established.”
>  ——The diagnostic ground truth is the real clinical diagnosis.
>
> "We set the batch size as 4, learning rate as 0.001, and alpha as 0.0005", how were these hyper parameters set?
> ——These hyper parameters are from our best model.
>
> "We transform these segmentation ground-truths into more precise and complete ones: the ground-truths with 3 labeled layers are transformed into equivalent ones with 6 discriminative labeled parts", how?
> —— As mentioned in  Section 4.1, only RNFL, GCC, and choroid are delineated by professionals. According to the manual delineation, 3 delineated layers equal to 6 labeled layers in OCT images: layer above RNFL, RNFL, GCC, layer between GCC and choroid, choroid, and layer below choroid (see Figure 1). To put it more clearly, an image has six layers, and the second, the third and the fifth are labeled by 1, 2, 3; then we re-label all the six layers by 1,2,3,4,5,6 from the first to the sixth layer.
>
> What do you mean by: "We use unified original OCT images to get the thickness vectors on the basis of the pre-trained model."?
> ——This is a long story. Unified original OCT images means all the OCT images have the same proportion to the real retina size. This process is necessary to guarantee the RNFL thickness in all the OCT images has the same proportion to the truth thickness in real world.
> Why on the basis of the pre-trained model? Because the OCT images with segmentation labels are scaled differently because of the OCT machine itself. So we use these OCT images with segmentation labels to train S-D net, especially the network for segmentation. Note that we cannot directly use the thickness from these rescaled images to make diagnosis. Thus we use unified original OCT images, extract their thickness vectors according to the pre-trained S-D net, and then use these thickness vectors to fine-tune the network for diagnosis.
>
> Why first train the S-D net end-to-end and then afterwards tune the D net?
> ——Explained as above.

---

> ### Comment · ~Jie_Wang3 · 2018-05-14
> **Reply to reviewer3 - 2**
>
> “The percentage improvements listed in section 4.4 are percentage points improvements and the improvements listed as 1 % are actually 0.1 pp. “
> ——Right. Thanks for your carefulness, and we will correct this typo.
>
> “I see no reason to list averages of accuracy, sensitivity and specificity.”
> —— Will remove the average as suggested.
>
> “The comparison SVM approach is not described in adequate detail.”
> —— SVM approach in paper is implemented in Python using sklearn toolkit, we adopt radial basis function (RBF) as the kernel, and set the penalty parameter of the error term as 0.8.
>
> “P7, "We note that the bias is difficult to update during training.", what is the point?”
> —— This is the fact we observed in the experiment. And we analyze this in P7, section 4.6 and suggest that the initialization of bias is very important.
>
> “How should the regions listed, "0:100, -400:-300, -300:-200, -200:-100 and -100:0" be interpreted and what was put instead of these values, as a constant thickness vector length of 1024 seems to be assumed. ”
> —— Yes, the thickness vector has a length of 1024, and regions covered are filled with zero (0). For the thickness vector, region 0:100 means position 0 to position 99, region -100:0 means position 924:1023, and region -200:-100 means position 824:923, etc.
>
> “An experiment is also conducted on including thickness measurements of other layers, but the specifics of how this is done could be more clear. E.g. is the thickness vector extended with the information of the extra layers?”
> —— Right. As said in paper, each pixel in the width with a thickness value - number of pixels in the height. Each layer has its own thickness information.

---

### Decision · Program_Chairs · 2018-05-15
**Paper27 Acceptance Decision**

Reject